# Chemical (Alkali) Burn-Induced Neurotrophic Keratitis Model in New Zealand Rabbit Investigated Using Medical Clinical Readouts and In Vivo Confocal Microscopy (IVCM)

**DOI:** 10.3390/cells13050379

**Published:** 2024-02-22

**Authors:** Mehak Vohra, Abha Gour, Jyoti Rajput, Bharti Sangwan, Monika Chauhan, Kartik Goel, Ajith Kamath, Umang Mathur, Arun Chandru, Virender Singh Sangwan, Tuhin Bhowmick, Anil Tiwari

**Affiliations:** 1Shroff-Pandorum Center for Ocular Regeneration, Dr. Shroff’s Charity Eye Hospital, New Delhi 110002, India; mehak.s@pandorumtechnologies.in (M.V.); abhagour@gmail.com (A.G.); jyoti.rajput@pandorumtechnologies.in (J.R.); bharti.sangwan@pandorumtechnologies.in (B.S.); monika.chouhan@pandorumtechnologies.in (M.C.); kartik.g@pandorumtechnologies.in (K.G.); umang@sceh.net (U.M.); drsangwan.lvpei@gmail.com (V.S.S.); 2Pandorum Technologies Pvt. Ltd., Bangalore 560100, India; avkamath@pandorumtechnologies.in (A.K.); arun@pandorumtechnologies.in (A.C.)

**Keywords:** neurotrophic keratitis, alkali burn, mild form, severe form, ophthalmic imaging

## Abstract

Purpose: Chemical eye injury is an acute emergency that can result in vision loss. Neurotrophic keratitis (NK) is the most common long-term manifestation of chemical injury. NK due to alkali burn affects ocular surface health and is one of its most common causes. Here, we established a rabbit model of corneal alkali burns to evaluate the severity of NK-associated changes. Material methods: Alkali burns were induced in NZ rabbits by treating the cornea with (i) a 5 mm circular filter paper soaked in 0.75 N NaOH for 10 s (Mild NK) and (ii) trephination using a guarded trephine (5 mm diameter and 150-micron depth), followed by alkali burn, with a 5 mm circular filter paper soaked in 0.75 N NaOH for 10 s (a severe form of NK). Immediately after, the cornea was rinsed with 10 mL of normal saline to remove traces of NaOH. Clinical features were evaluated on Day 0, Day 1, Day 7, Day 15, and Day 21 post-alkali burn using a slit lamp, Pentacam, and anterior segment optical coherence tomography (AS-OCT). NK-like changes in epithelium, sub-basal nerve plexus, and stroma were observed using in vivo confocal microscopy (IVCM), and corneal sensation were measured using an aesthesiometer post alkali injury. After 21 days, pro-inflammatory cytokines were evaluated for inflammation through ELISA. Results: Trephination followed by alkali burn resulted in the loss of epithelial layers (manifested using fluorescein stain), extensive edema, and increased corneal thickness (550 µm compared to 380 µm thickness of control) evaluated through AS-OCT and increased opacity score in alkali-treated rabbit (80 compared to 16 controls). IVCM images showed complete loss of nerve fibers, which failed to regenerate over 30 days, and loss of corneal sensation—conditions associated with NK. Cytokines evaluation of IL6, VEGF, and MMP9 indicated an increased angiogenic and pro-inflammatory milieu compared to the milder form of NK and the control. Discussion: Using clinical parameters, we demonstrated that the alkali-treated rabbit model depicts features of NK. Using IVCM in the NaOH burn animal model, we demonstrated a complete loss of nerve fibers with poor self-healing capability associated with sub-basal nerve degeneration and compromised corneal sensation. This pre-clinical rabbit model has implications for future pre-clinical research in neurotrophic keratitis.

## 1. Introduction

Neurotrophic keratitis is a rare degenerative eye disorder characterized by decreased corneal sensation, usually due to damage to the corneal nerve fibers. It can cause severe pain, decreased vision, and infection. Symptoms include redness and swelling of the cornea, blurred vision, eye pain, light sensitivity, and a watery discharge [1]. The prevalence of neurotrophic keratitis is estimated to be between 1 in 10,000 and 1 in 50,000 people. It is more common in elderly individuals and those with a history of eye trauma or certain medical conditions, such as diabetes and chronic inflammatory diseases [2]. It is also more common in certain parts of the world, such as South and Southeast Asia. The incidence of neurotrophic keratitis in the United States is estimated to be around 1.5 cases per 100,000 people [3]. In Europe, the incidence is estimated to be around 0.5 cases per 100,000 people. In India, the incidence is estimated to be around 5 cases per 100,000 people [4].

The evaluation of neurotrophic keratitis typically involves a combination of clinical examination, diagnostic imaging, and laboratory testing. Clinical examination typically includes an eye exam to look for signs of decreased corneal sensation, such as the inability to feel a cotton applicator or sensitivity to touch or light. Additionally, a slit lamp examination can be performed to look for signs of inflammation. Diagnostic imaging, such as corneal confocal microscopy, can be used to examine the cornea in greater detail [5]. It can help identify areas of nerve damage and measure corneal sensitivity. Laboratory tests, such as a tear film analysis, can be used to measure tear production and evaluate signs of infection. IL6 is a known pro-inflammatory cytokine and plays a very important role in persistent epithelial defect, which is a key feature of NK [6]. MMP9 is important during tissue remodeling, wound healing, and fibrosis [7]. Additionally, a lab test for neurotrophic keratitis can measure levels of nerve growth factor in the tears.

Currently, the primary treatment for neurotrophic keratitis is to reduce symptoms and limit further damage to the cornea. This includes using artificial tears, ointments, and steroid drops to reduce inflammation and lubricate the cornea [8]. Additionally, specific procedures such as tarsorrhaphy or amniotic membrane can be used to attempt to restore corneal epithelialization and limit further damage to the eye. In more severe cases, procedural interventions such as corneal neurotisation may be necessary [9,10,11,12]. The limitations of the current treatments for neurotrophic keratitis are that they are often only partially successful at restoring corneal sensation and may not be effective in preventing further damage to the eye. Additionally, the risk of infection increases with these treatments, and the potential for complications is higher [13]. In emergency cases, a corneal graft may be required, which further complicates corneal wound healing.

There are a few animal models of neurotrophic keratitis, including the rat, mouse, and rabbit models. These models are used to study the effects of neurotrophic keratitis on the cornea and evaluate potential treatments. The mouse model involves the injection of capsaicin into the cornea to induce a decrease in corneal sensation. In the rabbit model, various treatments are used to induce corneal neurotrophic keratitis, including contact lens wear, chemical ablation of corneal nerves, and chemical or mechanical stimulation of the cornea [14,15].

The current animal models of neurotrophic keratitis have several limitations. First, the models vary in the type of injury-induced and the severity of the corneal damage. Second, the models may not accurately represent the human condition, as corneal nerve damage in humans is often caused by a variety of factors, such as contact lens wear or eye trauma. Third, the models may not accurately represent the natural course of the disease in humans, as corneal nerve damage in humans is often progressive and worsens over time. Finally, the models may not be able to accurately assess the efficacy of potential treatments, as the treatments may vary in their efficacy based on the type and severity of the corneal nerve damage. Therefore, it is important to evaluate the clinical features of NK as per the human standard using imaging techniques closest possible to humans, making the translational aspect of the NK model possible.

In our current study, we generated two types of alkali burn-induced NK models: (i) direct burning of the rabbit cornea with alkali (mild form of NK) and (ii) trephination followed by alkali burn (severe form of NK). We characterized both models here through (i) clinical evaluation—decreased corneal sensation, pain, redness, and blurred vision; (ii) ophthalmic evaluation/imaging using OCT, slit lamp, pentacam, and IVCM; (iii) histology and inflammation. It can lead to serious complications such as corneal ulcers, scarring, and vision loss in severe cases.

## 2. Material and Methods

### 2.1. Study Design and Approval

The experimental design involved the use of Male New White Zealand (NWZ) rabbits that were subjected to ocular surface alkali burn in the right eye to study the development and progression of alkali injury in the cornea leading to NK. All the animals were 8–10 weeks of age and 1.8–2.0 kg body weight. The animals were followed up for a period of one month, and their ophthalmic and histopathological parameters were monitored during the course of the experiment. The study was approved by the Institutional Animal Ethics Committee. All surgeries were performed with controlled temperature and humidity under the supervision of the veterinarian and animal care team. All the animals were handled in agreement with the Association for Research in Vision and Ophthalmology (ARVO) statement for animal use in Ophthalmic and Vision Research.

### 2.2. Corneal Alkali Burn in Rabbits

Before the operation, the rabbits were anesthetized using ketamine hydrochloride (30 mg/kg) and xylazine (5 mg/kg) through intramuscular injection, and a topical drop of proparacaine was used in the right eye. A fresh 0.75 N NaOH solution was made, and a single circular Whatman filter paper with a diameter of 5 mm was soaked with 10 µL of 0.75 N sodium hydroxide (NaOH) solution.

Two kinds of injury models were made. For the first model **(mild form of NK)**, 10 µL of 0.75 N NaOH, soaked in 5 mm diameter Whatman paper for 10 s, was directly applied and washed with 10 mL normal saline for 30 s. For the second model **(severe form of NK)**, corneal tissue was excavated using a guarded trephine with a 5 mm diameter and 150 μm depth. A single circular filter paper with a diameter of 5 mm was cut, and 10 µL of 0.75 N sodium hydroxide (NaOH) solution was applied on the Whatman filter paper, subsequently on the trephined site for 10 s. After 10 s, the eyes were rinsed with 10 mL of normal saline to wash off excess alkali. The rabbit corneas from both models (N = 4 for the mild form of NK and N = 8 for the severe form of NK) were evaluated at regular intervals using ophthalmic imaging techniques. Antibiotic eye drops (Moxifloxacin) were administered in the eyes four times a day immediately after surgery for a minimum of 7–10 days until re-epithelization occurred. The maximal fibrotic response to epithelium-stroma post-alkali injury occurs at approximately 25–30 days after injury. Hence, the rabbit corneas were evaluated for 30 days.

### 2.3. Ophthalmic Imaging Techniques

After the ocular surface alkali injury, animals were followed up for the progression of the disease by obtaining clinical photographs at regular time intervals using the following techniques:

Slit Lamp: The rabbits were held in the correct position such that the entire frame of the cornea was visible, and images were acquired. Multiple images at 10X magnification were quickly taken to avoid eye drying using a slit-lamp biomicroscope (SL-CAM ICM 7, Appasamy Associates, Tamil Nadu, India) under different lighting conditions to examine corneal status and neovascularization. Diffuse light and oblique slit images indicated corneal clarity and a 3D view of the cornea, while fluorescein dye images taken with a blue light filter examined corneal abrasion by staining the underlying layer of collagen or hydrophilic stroma, which was exposed due to the corneal epithelium damage. Corneal neovascularisation was examined by acquiring the images in the green light filter.

Pentacam: Pentacam (Oculus, Wetzlar, Germany) is a rotating scheimpflug camera that helps us understand corneal biomechanics (Densitometry and Corneal Curvature) at different diameters of the cornea. This was a dark room procedure, so lights were switched off to image the rabbit cornea, and scans were taken from the center. The images were processed and analyzed for corneal curvature (keratometry values) and opacity (densitometry values) using built-in software (Pentacam HR, V1.21r65) in the machine.

Anterior Segment Optical Coherence Tomography (AS-OCT): AS-OCT is widely used to measure corneal thickness and epithelial and stromal hyperreflectivity. An anterior segment scan of the rabbit cornea was performed using the AS-OCT (Optovue, Avanti RTVUE-XR, Fremont, CA, USA). For imaging the cornea, the rabbits were held by a trained technician, an OCT probe was brought closer to the eye, and a scan was taken from the center. A single-trained optometrist obtained all images. The images were processed and analyzed for corneal thickness using the machine’s built-in software (Avanti XR, version 2016.1.0.26).

Aesthesiometer: Corneal sensitivity of the rabbit eyes was measured using a Cochet–Bonnet aesthesiometer (Luneau Technology, Pont-de-l’Arche, France). This instrument is typically used clinically to evaluate neurotrophic keratopathy. The handheld aesthesiometer (Cochet–Bonnet) contains a thin, retractable, nylon monofilament extending up to 60 mm in length. The device can apply variable pressure by adjusting the length. The monofilament ranges from 60 mm to 5 mm, and as the length is decreased, the pressure increases. We measured the corneal sensation of both eyes at the center of the cornea (within 5 mm diameter) and at the periphery (more than 5 mm diameter).

IVCM: In vivo, confocal microscopy (RCM 3, Heidelberg, Germany) of the cornea was performed on live animals in real-time. The high contrast of this laser-scanning system was critical for detecting delicate regenerating epithelial basal cells and nerves, while the high axial resolution enabled precise depth localization of individual repopulating keratocytes. The objective lens of the microscope was cleared using lint-free tissue, and a large drop of ophthalmic gel was applied on the front surface of the microscope objective lens. Tomocap was placed securely over the gel-covered objective lens, ensuring that the gel was flattened over the front surface of the objective lens and no air bubbles were present. Once the cornea was aligned with the tomocap, the images of different layers were acquired by adjusting the focal depths. For a full-thickness scan, the acquisition started at the superficial epithelium and was posteriorly adjusted to the focal depth. Post-acquisition, the images were analyzed for epithelial, keratocyte, and endothelium cellular density along the sub-basal nerve plexus.

### 2.4. Histological Analysis

After 30 days, rabbits were sacrificed with an overdose of ketamine and xylazine. Tissues of the cornea were collected from NK-diseased and healthy rabbits to evaluate the differences between the two forms of the NK model generation (i.e., Mild and Severe Form). The tissues were briefly washed with saline and then cut into two halves; the first half was used for histological analysis, and the second half was used for real time PCR.

For histology, sections were fixed in formalin, paraffin blocks were made, and sections were stained with Haematoxylin for 10 min and further differentiated with 1% acid alcohol for 30 s. They were further washed in running tap water for 10 min and then rinsed in 95% alcohol for 20 s. The sections were later counterstained with Eosin Y for 30 s, after which the sections were dehydrated in 95% alcohol and two changes of absolute alcohol for 5 min each. Sections were cleared with two changes of xylene for 5 min each and mounted with DPX (Dibutyl phthalate Polystyrene Xylene) mounting medium.

ELISA from Tear Samples: Schimer’s strip was used to collect the tear samples. 1x PBS with Protease Inhibitor Cocktail (PIC) was used to extract protein from the tear sample, which was further quantified using the BCA method. Commercially available kits of rabbit ELISA for MMP9, IL-6, and VEGF from CusaBio were used to estimate the level of metalloproteinase, pro-inflammatory cytokine, and angiogenesis factor from the tear samples of the healthy cornea and mild and severe form of NK models post 21 days of alkali injury.

Total RNA was isolated using the TriZol method (Invitrogen, Waltham, MA, USA) from the cornea of the control/healthy eye, mild form of NK, and severe form of NK. Complementary DNA was generated via reverse transcription reaction using the Revert Aid First Strand cDNA Synthesis Kit (Thermo Scientific, Waltham, MA, USA). Real-time quantitative polymerase chain reaction (PCR) was performed using SYBR Green Master Mix (Qiagen, Hilden, Germany) in the Azure Cielo 6 Dx real-time PCR machine (Azure Biosystems Inc., Dublin, CA, USA).

PCR reaction setup comprised 5 µL of 1:10 diluted cDNA, 7.5 µL of Power Track SYBR Green Master Mix (SYBR-Green; Applied Biosystems Life Technologies, Waltham, MA, USA), 1 µL of each primer with a concentration of 10 µM, and 1.5 µL of ddH20 in a total volume of 15 µL. Real-time PCR was performed with an initial denaturation step of 3 min at 95 °C, followed by 40 cycles of 10 s at 95 °C, 40 s at an annealing temperature of 60 °C, and 45 s at 72 °C. The relative quantitative expression levels of the alpha-SMA (Appendix A) were evaluated after normalizing to levels of GAPDH as endogenous control and compared to a normal cornea sample as a reference for the calculation of ∆∆Ct values, and fold change was calculated with the 2^−ΔΔCT^ method. The results are presented as significant increases or decreases in mRNA detected in the experimental groups (cornea of mild and severe form of NK) compared with their respective controls (healthy cornea). 

### 2.5. Statistical Analysis

Data were represented as mean values with standard error, obtained using Graph Pad Prism version 8.0.2. Appropriate non-parametric one-way ANOVA tests were used to obtain significance between multiple time points post-alkali burn. A comparison of two different groups was made using the Mann–Whitney test. The threshold for statistical significance was defined as *p* < 0.05.

## 3. Results

### 3.1. Corneas after Trephination Followed by Alkali Treatment Cannot Regenerate Compared to Direct Alkali Burn Revealed through Slit Lamp Evaluation

The ocular surface alkali burn was evaluated for a period of three weeks in both model types, and it was observed in both the models that there was an increase in cornea haze at day 0 and day 1 post alkali burn. Post one week, in the case of direct alkali burn, there were reduced signs of corneal scarring and haze, while the trend was reversed in the case of the severe form of NK, where we found an increase in the corneal haze post-day 7 onwards till day 21. In slit lamp diffuse light (Figure 1A,B, Diffuse Light), on day 15 and day 21, we can observe a significant difference in the level of corneal scarring in the mild vs. severe form of the NK disease model. Neovascularisation (Figure 1B, Green Light) was observed in the severe form of the NK model, unlike the mild form (Figure 1A). Because of the intact limbus, re-epithelization took place within 7–10 days in mild form, while it took slightly longer time in severe form, and the cornea healed with stromal scarring, unlike the mild form where there is a decreased level of corneal scarring and opacity. Hence, we found that the administration of 0.75 N NaOH for 10 s was sufficient to cause scarring, which tended to regenerate over time, while in the case of the severe NK model, associated with tissue excavation and alkali burn, fibrosis sets in as the disease progressed.

### 3.2. Higher Opacity Score in Severe Form of NK Models vs. Mild Form of Animal Model: Pentacam Evaluation

The Corneal Opacity Score was quantified using pentacam (Figure 2), where we found an immediate increase in the opacity score post alkali burn in the mild (Day 0, 0–2 mm: 59.13 ± 14.79 and 2–5 mm: 55.85 ± 6.63) and severe (Day 0, 0–2 mm: 56.04 ± 7.09 and 2–5 mm: 56.04 ± 7.09) form of the NK model, which was further increased on day 1. With disease progression, in the mild form of NK, we saw a trend of decreasing corneal opacity score from day 15 to day 21 (Day 15, 0–2 mm: 63.68 ± 10.30 and 2–5 mm: 46 ± 6.16; Day 21, 0–2 mm: 35.60 ± 3.47 and 2–5 mm: 35.13 ± 2.77). While in the severe form of the NK model, we saw a gradual increase in the corneal opacity from day 15 to day 21(Day 15, 0–2 mm: 81.50 ± 8.2, 2–5 mm and 67.39 ± 6.06; Day 21, 0–2 mm: 90.89 ± 3.77 and 2–5 mm: 73.25 ± 1.41) (Table 1) This indicates a transient form of corneal haze and opacity being formed in direct alkali burn compared to the severe form, where a more fibrous and opaque cornea with neo-vascularization develops.

### 3.3. Thicker and Hyper-Reflective Cornea in Severe Form of Alkali Burn: Assessed through AS-OCT

Pachymetry indicates corneal thickness, whereas hyperreflectivity is an indication of corneal opacity (the higher the hyperreflectivity, the more opaque the cornea is); both of these readouts are collected from AS-OCT. The pachymetry reading for a healthy rabbit cornea was 405.8 ± 14.21 µm thick. In Figure 3, upon ocular surface alkali burns using the direct method of application, corneal thickness increases due to swelling on day 0 and day 1 but as the inflammation subsides, we saw a gradual decrease in corneal thickness (central and peripheral) from day 7 (Central Thickness: 414.5 ± 92.09, Peripheral Thickness: 399.1 ± 67.17) to day 21 (Central Thickness: 341.8 µm ± 14.44, Peripheral thickness: 354 µm ± 11.88) (Figure 3A, Table 2). In the severe form of the NK model, a gradual increase in corneal thickness, both central and peripheral, from day 7 (Central Thickness: 476.7 µm ± 33.79, Peripheral thickness: 736.0 µm ± 22.74) to day 21 (Central Thickness: 485.5 µm ± 24.17, Peripheral thickness: 653 µm.5 ± 16.50) (Figure 3D) was observed due to underlying inflammation and corneal edema.

Hyperreflectivity is an indicator of opacity. An opaque cornea generates more red signals, whereas blue and green represent a more transparent cornea. As shown in Figure 3A, there is minimal distribution of red color and maximum presence of blue and green color in a healthy state before ocular injury, indicating transparent cornea. Post-burn, there is an increase in hyper-reflectivity, which is sustained till day 21 in the severe form, which is depicted by the extensive distribution of red color throughout the cornea, indicating the presence of opacity deep in the stroma close to the endothelium. In the case of direct burn, there is an increase in hyperreflectivity on day 0 and day 1, but subsequently, it decreases till day 21 and comes to a near-normal state.

### 3.4. Loss of Corneal Sensation and Cellular Density in the Severe Form of Animal Model, as Assessed through In Vivo Confocal Microscopy

In the severe forms of NK, corneal innervation by the trigeminal nerve is impaired. Partial or complete loss of corneal sensation may result in epithelial keratopathy, epithelial defect, stromal ulceration, and eventually corneal perforation. We see a decreased trend in epithelial and keratocyte density per mm^2^ (Figure 4 and Table 3), along with a loss of nerve fiber density and corneal sensation, both at the center and periphery, as evidenced by the decreased length of the nylon monofilament as compared to healthy rabbit corneas. In the mild form, we see a trend of decreased cellular density and nerve fiber density compared to healthy rabbit corneas, which is greater than in the severe form of NK. Thus, the readouts collected from IVCM and Cochet–Bonnet aesthesiometer indicate effects on sub-basal nerve plexuses to simulate NK development.

ELISA: The tear samples were analyzed for different soluble factors involved in inflammation and angiogenesis on day 21 post-alkali injury. We found the level of metalloproteinase MMP 9 (Figure 5A) to be higher in the mild form of NK compared to the severe form, while the levels of IL-6 and VEGF were reversed (Figure 5B,C). Hence, inflammation and angiogenesis were more prominent in the severe forms of NK compared to the mild form.

### 3.5. Histological Analysis

In the healthy rabbit cornea, the Corneal Epithelium was normal, with perfectly aligned cells of epithelium (squamous, wing, and basal cells) and a well-differentiated Bowman’s membrane. When compared between the two model systems, the epithelium was thicker compared to the severe form, with elongated basal cells that were not properly aligned, and Bowman’s membrane could not be demarcated as in a healthy cornea. When stroma was compared, we noticed a reduced number of keratocytes in the severe form of NK compared to the mild form. Descemet’s membrane and corneal endothelium were properly intact, with proper demarcation in the healthy, mild, and severe forms of NK. (Figure 5D–F). Treatment with alkali in two different forms, either direct burn or trephination followed by a burn, led to the formation of different grades of neurotrophic keratitis in rabbit corneas, as demonstrated by corneal thickness, transparency, epithelial cell, keratocytes, fibrosis, and nerve density and sensation, as shown in Figure 5G. In the mild form of NK (depicted by blue dotted lines), corneal thickness and epithelial and keratocyte density come to near normal over time, with reduced nerve sensation and the formation of a nebular scar, while the severe form of NK (depicted by orange dotted lines) represents a higher grade of disease where there is significant reduction in nerve fiber density and sensation from the healthy state and leucomatous scar develops with a thickened cornea. For observing the intensity of the scar, expression of alpha-SMA was evaluated in the rabbit cornea’s and we observed significantly higher expression of alpha-SMA in the severe form of NK as compared to the control and mild form (Appendix A). Thus, Figure 6 summarizes the study where inflammation and fibrosis play a key role in the generation of the disease model. 

## 4. Discussion

This is the first study, to the best of our knowledge, that demonstrates a severe form of NK model generation using a combinatorial approach of trephination and alkali burn compared to direct alkali burns for studying the underlying disease pathology of neurotrophic keratitis in rabbits. The most widely used laboratory animals for studying the model systems are mice, rats, guinea pigs, and rabbits. Among the reasons for the increasing use of rodents, such as mice, instead of rabbits are reduced maintenance costs, small size, availability of inbred strains, ease of breeding, short reproductive cycle, high numbers of progeny, and the wide availability of many knockouts (KO) and transgenic models [16]. However, rabbits have the advantage of an intermediate size between rodents and primates, longer life span than that of rodents, almost similar shape and size of the cornea as humans, and the immune system genes of rabbits are apparently more similar to those of the human immune system compared to rodent genes [17]. Hence, our study used rabbits as models to generate neurotrophic keratitis and, for the first time, examined many ophthalmic parameters in a single study in complete depth.

The severe form of the model system generated in rabbit corneas mimics grade 3 neurotrophic keratitis in humans with the clinical features of non-healing epithelial defect with surrounding edema, scalped margins of epithelial defects, reduced corneal sensation, stromal ulceration, and angiogenesis, while the mild form of NK depicts grade 1 neurotrophic keratitis, which is majorly associated with corneal haze [3]. Information gathered here from this study will help in the following two ways: first, for understanding the cascade of events leading to corneal scarring and loss of nerve sensation in neurotrophic keratitis; second, many emerging therapies targeting to prevent or reverse NK are being explored by various research groups. Thus, developing these animal models will also help in testing therapies for both their safety and efficacy [18,19].

In the current study, an array of ophthalmic and molecular parameters, like corneal thickness, opacification, nerve innervation, nerve sensation, and inflammation, was explored to investigate the detrimental effects of alkali burn in two model systems. With the aid of AS-OCT and Pentacam, we were able to demonstrate that both corneal thickness and opacity decrease over time in the direct alkali burn model, while a significant increase in thickness, opacity, persistent epithelial defect up to 15 days, and loss of corneal sensation occurs in the severe form of the NK model in rabbits. With the use of non-invasive techniques like in vivo confocal microscopy (IVCM) and Cochet–Bonnet aesthesiometer, we found that nerve sensation and the number of trigeminal nerves decreased significantly in the severe form of NK compared to the mild form. IVCM is a powerful instrument that clearly shows layer-by-layer changes occurring in the healthy and diseased cornea. As the results show, typically in the severe form of NK, the epithelial and stromal cell densities are significantly reduced along with no nerve fibers. The stroma is hyper-reflective with the presence of activated dendritic cells in the severe form compared to the mild form. As a consequence of corneal sensitivity loss or reduction, dry-eye disease can develop, causing neuropathic pain, corneal ulcers, and delayed corneal wound healing [20,21,22]. All these clinical features of the persistent epithelial defect and stromal ulceration were evident in the grade 3 model of NK in rabbits. Although these imaging tools have been around for more than 20 years, adapting them to detect and relate to clinical features has not been explored to the full extent. With this study, we are trying to demonstrate the power of this non-invasive imaging tool in assessing the microscopic pathology of the disease.

Additional clinical features, like corneal neovascularization, were observed as a late-stage result of chemical burns. Usually, the growth of new blood vessels is facilitated by increased expression of angiogenic cytokines, which were evaluated in the tears of rabbits. An increase in the levels of IL-6 and VEGF was observed in the severe form compared to the direct form, which clinically correlated with 360-degree angiogenesis captured in the diffuse light images of the slit lamp, suggesting underlying dynamic processes of inflammation-mediated angiogenesis taking place. Hence, when ocular inflammation occurs, corneal epithelial and endothelial cells, macrophages, and certain inflammatory cells produce angiogenic growth factors, namely VEGF and fibroblast growth factors. This inflammatory milieu, along with nerve damage created in the severe form of NK, leads to delayed corneal re-epithelization compared to the milder form, where we observe complete epithelization post-one-week, as no fluorescein stain was taken up by the rabbit cornea, unlike the severe form where we observed epithelial defect to be persistent till day 15 post alkali injury. In Herpetic stromal keratitis also, researchers have reported that proinflammatory cytokines such as IL-6 and TNF-α provoke angiogenesis through increased expression of VEGF [5,23]. IL-6 plays and modulates many important roles in ocular inflammation and angiogenesis in the conjunctiva, cornea, iris, retina, and orbit [24].

Apart from inflammation and angiogenesis, the level of matrix metalloproteinases (MMPs), which are zinc-dependent proteinases and play a key role in wound healing, were evaluated. In our study, we assessed the expression level of total MMP 9, which is a key metalloproteinase known to degrade extracellular matrix (ECM) proteins, thereby regulating fibrosis. We found levels of MMP9 to be decreased in the severe form compared to direct alkali burn. Our observation of decreased MMP9 in the severe form of NK is in line with Fini’s group observation, where MMP9 knockout showed delayed epithelial wound healing [6,7]. Henceforth, clinically relating it to the extent of opacity obtained in the two model systems, with an opaque cornea in trephination with alkali (a severe form of NK) vs. milder opacity in direct alkali burn. We also observed significantly increased expression of alpha-SMA at the transcript level in the severe NK form compared to the control and milder forms of NK (Appendix A). Thus, the data suggest more disorganized extracellular matrix deposition through myofibroblast in the severe form of NK than in the mild form, which produces corneal haze.

In 1982, Schimmelpfennig and R. Beuerman gave a very primitive rabbit animal model of NK using thermocoagulation of the ophthalmic branch of the trigeminal nerve [25]. This study highlighted the use of a slit lamp, histology, and corneal sensation to establish the base for the animal model [25]; however, it failed to comment on the corneal opacification, vascularisation, and other cellular changes taking place in the rabbit cornea and are key features to relate it clinically in humans. Attempts have been made to make the model in other animal species, like mice and rats, using either thermocoagulation or capsaicin injection at the neonatal stage [14]. Unfortunately, the injection of capsaicin at the neonatal stage does not mimic the clinical cause of neurotrophic keratitis in humans.

In summary, this study describes the development of a reliable rabbit animal model of neurotrophic keratitis of both grade 1 and grade 3 of the disease. We evaluated in-vivo longitudinal corneal metrics of swelling, opacification, nerve sensation, and neovascularization resulting from alkali burns in two model systems (Figure 6) and found trephination followed by alkali burn proved to be a good, stable, and reproducible model that closely resembles the clinical features of NK in human eyes as per the clinician. Hence, this NK model can be used for testing both prophylactic and therapeutic modalities in the near future.

## Figures and Tables

**Figure 1 cells-13-00379-f001:**
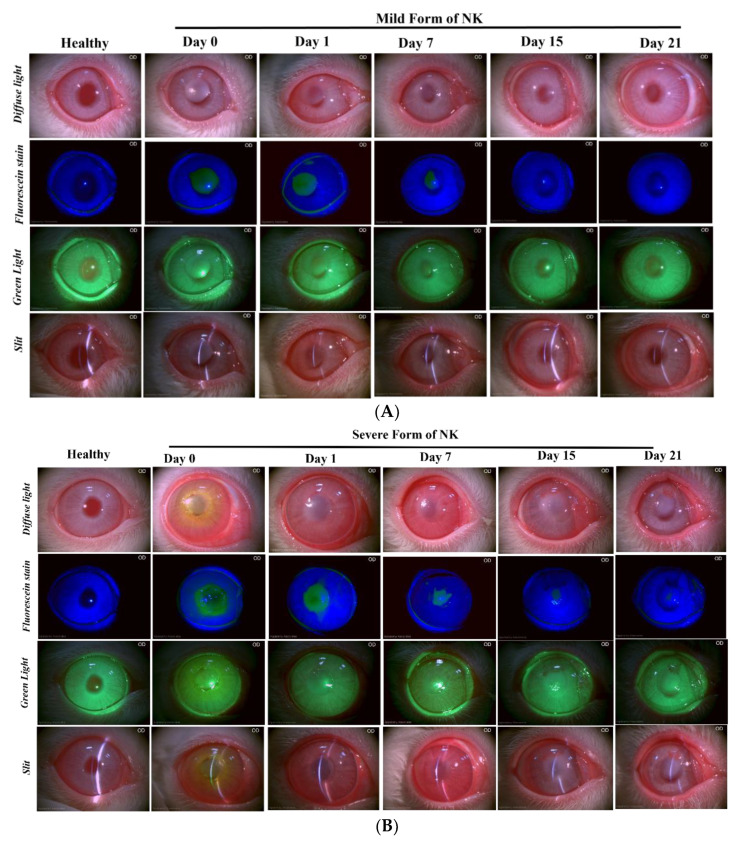
**Slit Lamp examination of the healthy, mild, and severe forms of the NK model post alkali burn at different time points.** Diffuse light indicates cornea clarity, fluorescein stain indicates an epithelial defect in the blue filter, the green light indicates the level of neovascularization, and the slit indicates the 3D view of the cornea from the mild (**A**) and severe (**B**) forms of the NK model.

**Figure 2 cells-13-00379-f002:**
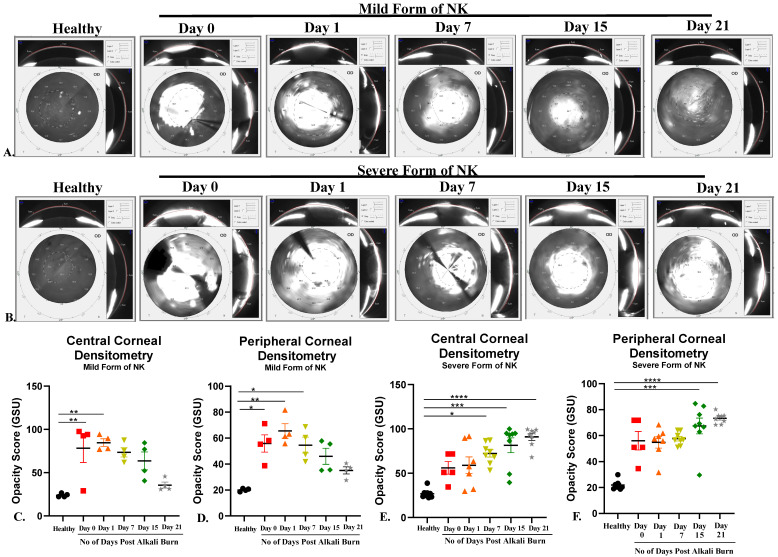
Opacity score (an indicator of transparency) obtained from the healthy, mild, and severe forms of the NK disease model using Pentacam. (**A**,**B**) Corresponding representative images showing opacity scores in healthy, mild and severe NK disease models. (**C,D**) Bar graphs represent quantified data at both the center (0–2 mm) and periphery (2–5 mm) at different time points from the mild form of the NK disease model. (**E**,**F**) Bar graphs represent quantified data at both the center (0–2 mm) and periphery (2–5 mm) at different time points from the severe form of the NK disease model. N = 4 for the mild form of NK and N = 8 for the severe form of NK. The * represents statistically significant data (*p*-value < 0.05) between various study arms (* means *p*-value < 0.05, ** < 0.005, *** < 0.0005 and **** < 0.0001).

**Figure 3 cells-13-00379-f003:**
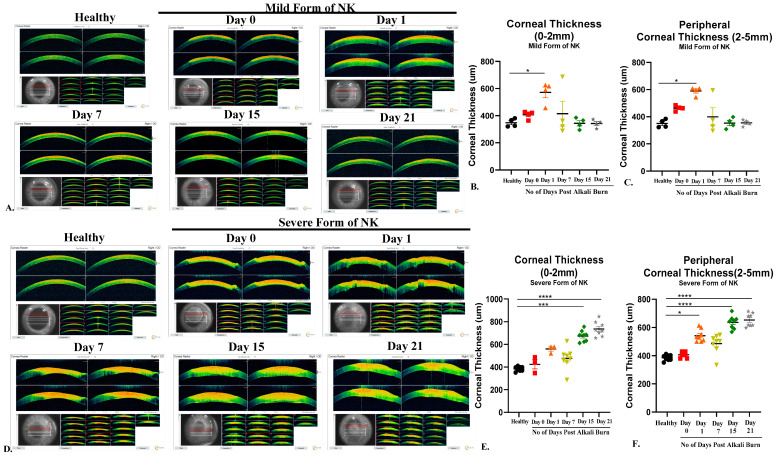
Pachymetry (Corneal thickness and hyper-reflectivity) in the healthy, mild, and severe forms of the NK model at different time points. (**A**) Corresponding representative images showing corneal thickness in the healthy and mild forms of NK model generation at different time points. (**B**,**C**) Bar graphs represent statistically significant differences in central (0–2 mm) and peripheral (2–7 mm) corneal thickness of ocular surface post alkali burn in the mild form of the NK disease model at different time points. (**D**) Corresponding representative images showing corneal thickness in the healthy and severe forms of NK model generation at different time points. (**E**,**F**) Bar graphs represent statistically significant differences in central (0–2 mm) and peripheral (2–7 mm) corneal thickness of ocular surface post alkali burn in the severe form of the NK disease model at different time points. N = 4 for the mild form of NK and N = 8 for the severe form of NK. The * represents statistically significant data (*p*-value < 0.05) among various study arms (* means *p*-value < 0.05, *** < 0.0005, and **** < 0.0001).

**Figure 4 cells-13-00379-f004:**
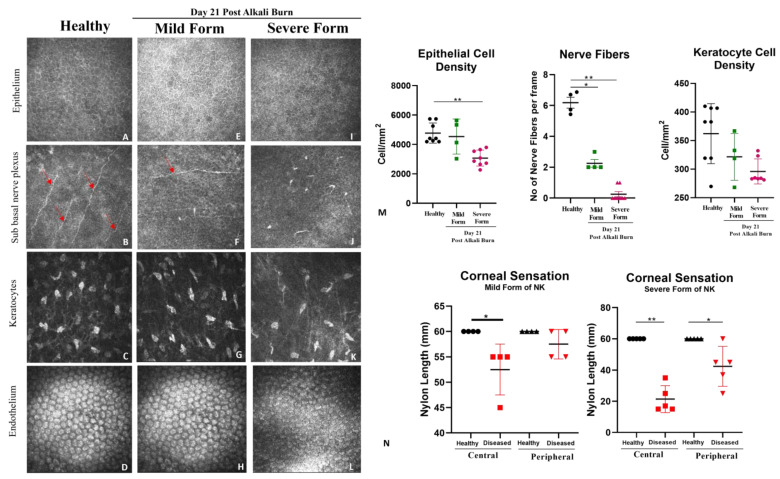
**Representative images from in vivo confocal microscopy conducted on live animals.** (**A**–**L**) Corresponding representative images showing epithelium, sub-basal nerve fibers (indicated by red arrows), keratocytes, and endothelium in healthy, mild and severe form of the diseased corneas. (**M**) Bar graphs represent quantified data for cellular density of corneal epithelium, keratocyte cells per mm^2^, and density of sub-basal nerve fibers in the healthy and diseased cornea (day 21 post-alkali injury, mild vs. severe). (**N**) Bar graph represents quantified data for corneal sensation in healthy, mild and severe form of NK both at the center and periphery. N = 4 for the mild form of NK and N = 8 for the severe form of NK. The * represents statistically significant data (*p*-value ≤ 0.05) between the two study arms (* means *p*-value < 0.05, ** < 0.005).

**Figure 5 cells-13-00379-f005:**
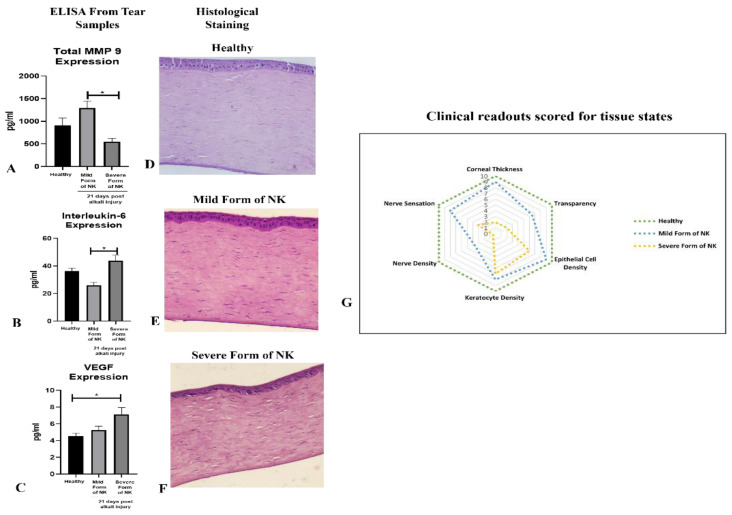
**ELISA and Histological Analysis of the healthy, mild, and severe forms of the NK model.** (**A**–**C**) ELISA of Metalloproteinase (MMP9), pro-inflammatory cytokine (IL-6), and angiogenesis factor (VEGF) from tear samples collected from the healthy cornea, mild, and severe forms of the NK model was performed post 21 days of alkali injury. N = 4 for the mild form of NK and N = 8 for the severe form of NK. The * represents statistically significant data (*p*-value ≤ 0.05) between the two study arms (* means *p*-value < 0.05). (**D**–**F**) Representative hematoxylin and eosin histopathology images from the healthy cornea, mild, and severe forms of NK model. (**G**) Radar plot representation of corneal tissue states derived from pachymetry, densitometry, aesthesiometer, and IVCM data obtained from healthy rabbits and diseased forms (mild and severe forms of NK) post 21 days of alkali injury. Corneal thickness, transparency, epithelial cell, nerve, keratocyte density, and nerve sensations are represented here on a relative scale of 0–10, with the scheme of conversion from the original machine readout to relative scoring provided in Appendix A.

**Figure 6 cells-13-00379-f006:**
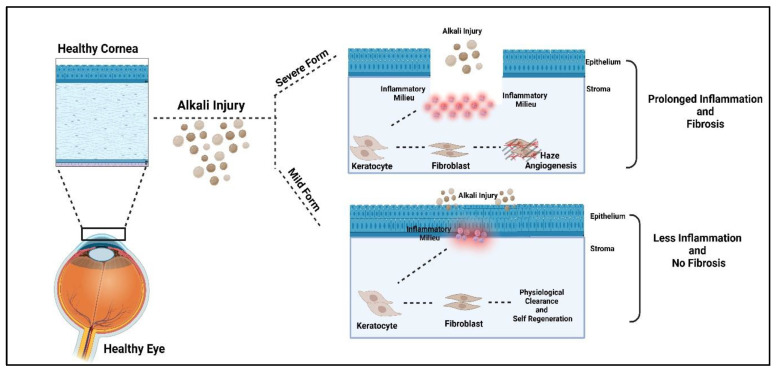
**Graphical and Functional Representation of the Mild and Severe form of NK model generation.** The study delineates the role of inflammation and fibrosis as the underlying mechanisms for the generation of the model. In the mild form, milder inflammation allows for physiological clearance and regeneration, while in the case of severe form of NK, exacerbated inflammation and fibrosis lead to clinical manifestations of scarring and angiogenesis.

**Table 1 cells-13-00379-t001:** Opacity score at central and peripheral diameters of the cornea at different time points in the healthy, mild, and severe forms of the NK disease model. The * represents statistically significant data (*p*-value < 0.05) between various study arms (* means *p*-value < 0.05, ** < 0.005, *** < 0.005, and **** < 0.0001).

Central Corneal Densitometry (0–2 mm, In GSU Unit, Mean ± SEM): Mild Form of NK Model					
Healthy	Day 0	Day 1	Day 7	Day 15	Day 21
23.95 ± 1.03	59.13 ± 14.79 **	84.73 ± 4.28 **	73.58 ± 5.41	63.68 ± 10.30	35.60 ± 3.47
**Central Corneal Densitometry (0–2 mm, In GSU Unit, Mean ± SEM): Severe Form of NK Model**					
26.98 ± 1.84	56.04 ± 7.09	59.06 ± 9.3	72.43 ± 4.13 *	81.50 ± 8.2 ***	90.89 ± 3.77 ****
**Peripheral Corneal Densitometry (2–5 mm, In GSU Unit, Mean ± SEM): Mild Form of NK Model**					
20.08 ± 0.57	55.85 ± 6.63 *	65.45 ± 5.6 **	54.63 ± 6.08 *	46 ± 6.16	35.13 ± 2.77
**Peripheral Corneal Densitometry (2–5 mm, In GSU Unit, Mean ± SEM): Severe Form of NK Model**					
21.94 ± 1.26	56.04 ± 7.09	54.84 ± 4.40	57.64 ± 1.7	67.39 ± 6.06 ***	73.25 ± 1.41 ****

**Table 2 cells-13-00379-t002:** Corneal thickness at central and peripheral diameters of the cornea at different time points in the healthy, mild, and severe forms of the NK disease model. The * represents statistically significant data (*p*-value ≤ 0.05) between various study arms (* means *p*-value < 0.05, *** < 0.0005, and **** < 0.0001).

Corneal Thickness (0–2 mm, In Micron, Mean ± SEM): Mild Form of NK Model					
Healthy	Day 0	Day 1	Day 7	Day 15	Day 21
347.8 ± 14.89	405.8 ± 14.21	571.8 ± 39.35 *	414.5 ± 92.09	343.5 ± 20.04	341.8 ± 14.44
**Corneal Thickness (0–2 mm): Severe Form of NK Model**					
383.5 ± 6.99	424 ± 40.63	559.7 ± 19.33	476.7 ± 33.79	674.0 ± 17.44 ***	736.0 ± 22.74 ****
**Corneal Thickness (2–5 mm): Mild Form of NK Model**					
350 ± 15	462.8 ± 9.47	589.4 ± 15.49 *	399.1 ± 67.17	352.9 ± 19.27	354 ± 11.88
**Corneal Thickness (2–5 mm): Severe Form of NK Model**					
385.4 ± 7.17	407.1 ± 6.78	542.2 ± 15.47 *	485.5 ± 24.17	636.8 ± 16.66 ****	653.5 ± 16.50 ****

**Table 3 cells-13-00379-t003:** Epithelial, keratocyte cellular density, and nerve fiber quantification using IVCM at Day 21 post alkali burn in the healthy, mild, and severe forms of the NK disease model. The * represents statistically significant data (*p*-value ≤ 0.05) between various study arms (* means *p*-value <0.05, ** < 0.005).

Cell Density/Nerve Fiber Density Per Frame (Mean ± SEM)	Healthy	Mild Form of NK	Severe Form of NK
Epithelial Density (Per mm^2^)	4773 ± 242.1	4536 ± 597.6	3072 ± 192 **
Nerve Fibers (Per frame)	6.18 ± 0.35	2.25 ± 0.25 *	0.25 ± 0.16 **
Keratocyte Density (Per mm^2^)	362.1 ± 18.57	321.5 ± 20.48	296 ± 8.28

## Data Availability

No new data were created or analyzed in this study. Data sharing is not applicable to this article.

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
