# Peer review of "Chemical (Alkali) Burn-Induced Neurotrophic Keratitis Model in New Zealand Rabbit Investigated Using Medical Clinical Readouts and In Vivo Confocal Microscopy (IVCM)"

_cells, 2024, doi:10.3390/cells13050379_

Round 1

Reviewer 1 Report

Comments and Suggestions for Authors

Neurotrophic keratitis (NK) due to alkali burn affects ocular surface health and is one of its most common causes. The authors established a rabbit model of corneal alkali burn to evaluate NK-associated changes severity. Alkali burns were induced by treating the cornea with 5 mm circular paper soaked in 0.75 N NaOH for 10 secs. Clinical features were evaluated by slit lamp, Pentacam, AS-OCT and NK-like changes, epithelial defects, subbasal nerve plexus, stroma by IVCM, and corneal sensation by aesthesiometer post-injury for 30 days. Post 30 days several markers (MMP-9, IL-6 and VEGF) were evaluated in the tears. The burn resulted in the loss of epithelium, extensive edema, and increased corneal thickness (550 um compared to 380 um thickness of control) and opacity score. IVCM showed complete loss of nerve fibers that did not regenerate over 30 days, and loss of corneal sensation, conditions associated with NK. Factors related to angiogenesis and inflammation were increased. The authors conclude that clinically, the alkali-treated rabbit model depicts the features of NK. This pre-clinical rabbit model has implications for future research in neurotrophic keratitis.

1. The abstract needs rewriting. 1. Please spell out the methods as the general reader is not familiar with specific terminology. This concerns AS-OCT and IVCM. 2. The authors need to enumerate significantly changed tear factors in the abstract and indicate whether they reverted to normal or not during follow-up. 3. The abstract should also mention that there was a complete re-epithelialization during the 30 days.4. The abstract does not mention two models used, only the milder one. This is important as most studied parameters markedly differed between two models.

2. Please justify the choice of tear factors studied and provide references.

3. The first subheading of results needs a change (trephination cannot regenerate).

4. " Hence, we found administration of 0.75N NaOH for 10 seconds was sufficient to cause scarring in the epithelium and anterior part of the stroma which tends to regenerate over time..." Please remove or explain with a picture what "scarring in the epithelium" could mean.

5. Tables should be statistically analyzed.

6. Fini's group has shown Knockout of MMP-9 slowed epithelial wound healing. A decrease in this enzyme might be responsible for the observed delay in re-epithelialization in the severe model.

7. The authors might like to assess scarring on corneal section by alpha-SMA staining.

8. Please add the number of animals in the figure and table legends.

Comments on the Quality of English Language

The authors may need to run a spell check for infrequent typos and avoid rarely used words like opaquer (common use more opaque).

Author Response

Thank you very much for taking the time to review this manuscript. Please find the detailed responses below and the corresponding revisions/corrections highlighted/in track changes in the re-submitted files.
  1. The abstract needs rewriting. 1. Please spell out the methods as the general reader is not familiar with specific terminology. This concerns AS-OCT and IVCM. 2. The authors need to enumerate significantly changed tear factors in the abstract and indicate whether they reverted to normal or not during follow-up. 3. The abstract should also mention a complete re-epithelialization during the 30 days.4. The abstract does not mention two models used, only the milder one. This is important as most studied parameters markedly differed between the two models.

Reply:  Authors thank the reviewer for the suggestions. As suggested we have modified the abstracts and incorporated the suggestions in the revised manuscript.

  1. Please justify the choice of tear factors studied and provide references.

Reply: IL6 is a known pro-inflammatory cytokine, IL6 plays a very important role in persistent epithelial defect, which is a key feature (PMID: 36839533), based on this information we selected IL6. MMP9 is important during tissue remodeling, wound healing, and fibrosis (PMID: 11689563). Both the references are incorporated in the revised manuscript.

  1. The first subheading of results needs a change (trephination cannot regenerate).

Reply: We thank the reviewer for the suggestion. We have incorporated the suggestion in the revised manuscript.

  1. " Hence, we found administration of 0.75N NaOH for 10 seconds was sufficient to cause scarring in the epithelium and anterior part of the stroma which tends to regenerate over time..." Please remove or explain with a picture what "scarring in the epithelium" could mean.

Reply: As suggested by the reviewer we have removed the word “scarring in the epithelium” in the revised manuscript.

  1. Tables should be statistically analyzed.

Reply: We thank the reviewer for the suggestions, and we have incorporated the suggestions in the revised manuscript.

  1. Fini's group has shown Knockout of MMP-9 slowed epithelial wound healing. A decrease in this enzyme might be responsible for the observed delay in re-epithelialization in the severe model.

Reply: We thank the reviewer for the great insight and we completely agree with the reviewer's suggestion. We will also incorporate Fini’s group paper as a reference (PMID: 11689563) to strengthen our discussion.

  1. The authors might like to assess scarring on a corneal section by alpha-SMA staining.

Reply: We thank the reviewer for the suggestions. In our present manuscript, we have focussed on generating neurotrophic keratitis (NK) and characterizing it by medical and clinical readouts and IVCM. Taking the reviewer's suggestion we evaluated alpha-SMA by real-time PCR (supplementary Fig 1.). We observed a significantly higher expression of alpha-SMA in the severe form of NK (20 fold) compared to milder and control. We plan to further characterize at the cellular and molecular level as a separate project.

  1. Please add the number of animals in the figure and table legends.

Reply: We agree with the reviewer and we have incorporated the suggestion in the revised manuscript.

Reviewer 2 Report

Comments and Suggestions for Authors

The authors performed in vivo analysis of rabbit corneas after alkali injury. However, couple additional experiments should be done:

1. The nerve fiber observed by confocal should be confirmed by immunostaining.

2. It is interesting that IL-6 expression in mild injury tear is less than healthy tear film. Similarly, MMP9 expression in severe injured tear is lower than that in healthy tear. Can the authors explain these observations?

Author Response

The authors performed in vivo analysis of rabbit corneas after alkali injury. However, a couple of additional experiments should be done:

  1. The nerve fiber observed by confocal should be confirmed by immunostaining.

Reply: We thank the reviewer for the suggestions. We would like to highlight that we are focussing on the medical clinical readouts and in vivo confocal imaging in the current study. Clinically NK is diagnosed by clinical evaluation of slit-lamp and sensation by aesthesiometer. Moreover, for nerve fiber staining it is preferred to have whole mount staining, for that we have to re-perform the entire rabbit experiment, which will take a lot of time. Our next project is focussed on cellular and molecular characterization which involves the immunostaining for the nerve fibre.

  1. It is interesting that IL-6 expression in mild injury tears is less than healthy tear film. Similarly, MMP9 expression in severely injured tears is lower than that in healthy tears. Can the authors explain these observations?

Reply: We thank the reviewer for the insights. Though we see that the IL6 is lower in mild injury tears compared to the control, but it was not statistically significant. Agreeing, with the observations our next project involves evaluating IL6 in a larger sample size and its signaling. Dr Fini’s group had shown that MMP9 knockout animals showed delayed epithelial wound healing, our observation is in line with the previously published literature (PMID: 11689563).

Round 2

Reviewer 1 Report

Comments and Suggestions for Authors

The comments have been generally well addressed. Please address some remaining minor concerns.

1. Statistics. Please change everywhere P ≤ ... to P < ...

2. The first subheading of results is still awkward. Please change it to "Corneas after trephination followed by alkali treatment cannot regenerate as compared to direct alkali burn revealed by slit lamp evaluation".

3. The justification of cytokine use presented in the rebuttal letter should be incorporated into the Introduction with pertinent references.

4. Please reference figure S1 in the methods (QRT-PCR) and the text.

Author Response

We thank the reviewer for the comments. It has significantly improved the quality of the manuscript. 

Please find attached the file for your considerations. 

Reviewer 2 Report

Comments and Suggestions for Authors

The authors have answered my questions.

Author Response

We thank the reviewer for reviewing the manuscripts and the constructive suggestions.